# *Helicobacter pylori* and Its Role in Gastric Cancer

**DOI:** 10.3390/microorganisms11051312

**Published:** 2023-05-17

**Authors:** Victor E. Reyes

**Affiliations:** Department of Pediatrics and Microbiology & Immunology, University of Texas Medical Branch, 301 University Blvd., Galveston, TX 77555-0372, USA; vreyes@utmb.edu; Tel.: +1-409-772-3824

**Keywords:** gastric cancer, *Helicobacter pylori*, cag pathogenicity island, cytotoxin-associated gene A, oncoprotein, vacuolating toxin A, immune evasion

## Abstract

Gastric cancer is a challenging public health concern worldwide and remains a leading cause of cancer-related mortality. The primary risk factor implicated in gastric cancer development is infection with *Helicobacter pylori*. *H. pylori* induces chronic inflammation affecting the gastric epithelium, which can lead to DNA damage and the promotion of precancerous lesions. Disease manifestations associated with *H. pylori* are attributed to virulence factors with multiple activities, and its capacity to subvert host immunity. One of the most significant *H. pylori* virulence determinants is the *cagPAI* gene cluster, which encodes a type IV secretion system and the CagA toxin. This secretion system allows *H. pylori* to inject the CagA oncoprotein into host cells, causing multiple cellular perturbations. Despite the high prevalence of *H. pylori* infection, only a small percentage of affected individuals develop significant clinical outcomes, while most remain asymptomatic. Therefore, understanding how *H. pylori* triggers carcinogenesis and its immune evasion mechanisms is critical in preventing gastric cancer and mitigating the burden of this life-threatening disease. This review aims to provide an overview of our current understanding of *H. pylori* infection, its association with gastric cancer and other gastric diseases, and how it subverts the host immune system to establish persistent infection.

## 1. Introduction

Gastric cancer (GC) remains among the deadliest cancers worldwide, taking close to one million lives annually. GC is the fourth leading cause of cancer-related deaths [1]. According to the American Cancer Society’s (ACS) estimates for 2022, in the United States, there are 26,380 new cases and 11,090 deaths attributed to GC [2]. Worldwide, in 2020 the incidence of GC was 1,089,103 and claimed 768,793 lives [3]. Although its incidence in the United States is declining, the prognosis for patients with GC is bleak since their five-year survival rate is low. The poor prognosis for these patients is because most GC cases (90%) are diagnosed at an advanced stage due to our inadequate understanding of the underlying mechanisms that regulate carcinogenesis and the lack of routine screening for GC.

*H. pylori* was initially described by Australian scientists Barry Marshall and Robin Warren in 1982 [4]. Their work with human gastric specimens identified *H. pylori* as the primary cause of chronic gastritis and peptic ulcer disease (PUD). Their research challenged the prevailing medical dogma that ulcers result from stress and lifestyle factors. Before their discovery, they treated peptic ulcers primarily using antacids and diet modification, such as avoiding spicy and acidic foods. However, these treatments were inadequate, and often vagotomy or surgery to remove the affected portion of the stomach or duodenum was the next step to reduce acid secretion. The work of Marshall and Warren transformed the treatment of PUD and has helped save many lives. Their efforts were recognized with the Nobel Prize in Physiology or Medicine in 2005. The infection is treated with antibiotics to eradicate the bacteria and acid-suppressing drugs.

The primary risk factor for GC is *Helicobacter pylori*, a microaerophilic, spiral-shaped Gram-negative bacterium classified as a class I carcinogen [5]. *H. pylori* is perhaps the most successful human pathogen since its infection is prevalent, infecting approximately one-half of the world’s population, or approximately 4.4 billion individuals [6]. The human stomach is the only natural reservoir of *H. pylori*. In developing countries, the prevalence may reach as high as 90% [6]. In developed countries with the lowest infection rates, up to 40% of all adults are infected with *H. pylori*. Infection is generally acquired during childhood and, without appropriate treatment, can last the individual’s lifetime. Limiting infection is practically impossible due to antibiotic resistance and elevated reinfection rates. Although *H. pylori* infection is strongly associated with GC, only 1–3% of infected individuals develop GC, while most remain asymptomatic. It is not clear why only some *H. pylori*-infected individuals develop GC. However, various factors likely influence the outcome of the infection, including host susceptibility, environmental factors, and virulence of the infecting strain.

## 2. *H. pylori* and Associated Diseases

### 2.1. Gastric Cancer

Although most persons infected with *H. pylori* are asymptomatic, infection with *H. pylori* may lead to several clinically significant disorders, including chronic gastritis, PUD, mucosal tissue-associated lymphoma, and GC. Chronic gastritis, if left untreated, may progress to atrophic gastritis and intestinal metaplasia, precancerous lesions in the sequence proposed by Pelayo Correa [7]. According to his model, *H. pylori* infection activates a cascade of histologic changes that progress through several phases, from chronic superficial gastritis to atrophic gastritis, intestinal metaplasia, and dysplasia, before ultimately resulting in GC. This model of GC progression stresses the importance of early detection and treatment of chronic gastritis, specifically in persons with a history of *H. pylori* infection. 

The most common type of GC is adenocarcinoma, which represents 95% of cases. GC is usually regarded as one condition and is classified based on the anatomic location within the stomach, histologic type of cancer, and stage [8]. Anatomically, GC is classified as either cardia/proximal (upper part of the stomach) or noncardia/distal (antrum and pylorus). Histologically, it may be classified as either diffuse or intestinal. The GC stage is determined by the tumor size, spread to sentinel lymph nodes, or whether it has metastasized to distant anatomical sites, such as the liver, lungs, or bones. Approximately 90% of noncardia GC cases are due to *H. pylori* infection [9]. Because of the wealth of epidemiologic data linking *H. pylori* in the development of GC, in addition to observations in animal models, *H. pylori* was classified as a class I carcinogen by the International Agency for Research on Cancer (IARC) together with the World Health Organization (WHO) [10].

The incidence of GC varies geographically across different parts of the world. It is highest in Eastern Europe, Eastern Asia, and Latin America, while the lowest incidence is in North America, Western Europe, Australia, and Africa [11]. Not surprisingly, the highest incidence of GC occurs in developing countries, coinciding with high *H. pylori* carriage rates. An interesting observation was made in patients in Africa, where the prevalence of *H. pylori* infection is very high, but the incidence of GC is low. This apparent discrepancy between the high prevalence of *H. pylori* infection in Africa and the low incidence of GC in these regions was initially referred to as the “African enigma” [12,13]. To address this paradox, Fox and colleagues investigated the possibility that concurrent parasitic infection could influence the immune response to *H. pylori*, initially reported by us and others as skewed toward a Th1 immune response [14]. They hypothesized that helminth infections, which stimulate Th2-polarized responses, could modify the Th1 immune response induced by *H. pylori* and thus change the outcome of the infection. They studied mice infected with *H. felis* with and without simultaneous infection with the enteric nematode *Heligmosomoides polygyrus*, which has a strictly enteric life cycle [15]. They observed that the nematode infection prevented the development of gastric atrophy. This correlated with a significant decrease in mRNA for cytokines and chemokines associated with a gastric inflammatory response of Th1 cells [15]. These observations led to the conclusion that nematode infections ameliorate gastric atrophy, a precancerous lesion. However, subsequent studies in mice by the same group suggested that concomitant infection with other *Helicobacter* species could differentially affect gastric pathology [16]. They noted that *H. muridarum* coinfection significantly attenuated *H. pylori*-associated gastric pathology; however, coinfection with *H. hepaticus* promoted *H. pylori*-associated gastric disease [16]. Interestingly, the exacerbated pathology in mice coinfected with *H. pylori* and *H. hepaticus* was not due to increased Th1 responses, since those mice had lower mRNA levels of gastric Th1 cytokines TNF-α, IFN-γ, and IL-1β than mice infected only with *H. pylori*. Instead, the dually infected mice had higher mRNA levels of gastric IL-17A than mice infected with *H. pylori* alone [16]. 

The concept of the “African enigma” was challenged by a study of the literature regarding PUD in the African continent [13]. As explained below, PUD is also linked to *H. pylori* infection, and similar observations have been made regarding its incidence in African populations; thus, PUD was considered a surrogate to examine its incidence in the context of *H. pylori* infection. The study concluded that the “African enigma” reflected inadequate data obtained from people lacking resources, healthcare access, and a comparatively short life expectancy [13]. However, a different set of *H. pylori*-related observations had been made with Asian populations with comparatively lower infection rates. One study reported the seroprevalence of *H. pylori* among Japanese and Chinese adults of approximately 50%, but the prevalence of GC in those populations is high [17]. One explanation for this higher prevalence of GC in East Asia could be differences in the *H. pylori* CagA virulence factor in East Asian strains compared to Western strains [18,19], which make those strains more virulent, as discussed in detail below. A more in-depth study to attempt to explain regional differences in GC prevalence in populations with similar rates of *H. pylori* infection was led by Correa’s team of investigators who examined GC rates among inhabitants in the state of Nariño, Colombia. The rates of GC among inhabitants of the high-altitude Andes Mountains were high (∼150 per 100,000), while the incidence rate of GC for those living at sea level was low (∼6 per 100,000) [20]. It is worth noting that the high-risk mountain (Tuquerres) population is only ~150 Mi away from the coastal low-risk populations (Tumaco). Although the prevalence of *H. pylori* infection is high in both groups (>80% after age 10), there were significant differences in how their inhabitants were affected. Atrophy was more common, and the incidence of GC was higher in high- versus low-altitude regions [20,21]. One study reported differences in the virulence genotypes (*cagA* positive, *vacA* s1, and m1) in both regions’ prevailing *H. pylori* strains [22]. It is important to note the differences between the inhabitants of those two regions. They differ in their ancestry, with primarily African origin in the coastal region (58%), and mostly Amerindian ancestry in the mountain region (67%) [23]. Additionally, dietary differences, the incidence of helminthiasis and toxoplasmosis, and, more recently, gastric microbiomes were reported to differ between both groups. Altogether, these observations in different regions led to considering the role of virulence of the infecting strains, the gastric microbiome, coinfections, environmental factors such as diet, and host genetics as factors influencing the outcome of the infection. 

### 2.2. Peptic Ulcer Disease

PUD is another condition that most often involves *H. pylori*, accounting for 90–95% of duodenal ulcers and 70–85% of gastric ulcers [24]; the remainder of cases are due to nonsteroidal anti-inflammatory drugs (NSAIDs). PUD is frequently defined as a rupture in the gastric or duodenal mucosa greater than 3–5 mm caused by an imbalance in mucosal protective and injurious factors [25]. PUD may have significant complications such as bleeding, perforation, penetration into adjacent organs, and obstructions, and death may result from these complications. A systematic literature review reported an average mortality of 8.6% after bleeding and 23.5% after perforation 30 days later [26]. Mortality increases with age, comorbidities, shock, and treatment delays [26]. Although the incidence of PUD has decreased recently due to improved diagnosis and treatment of *H. pylori* infection, PUD still is a public health issue that causes significant distress and severe complications if left untreated. The annual incidence of PUD is 0.1–0.3%, affecting approximately 10% of the population worldwide [27]. *H. pylori* infection and the ensuing inflammation can alter the gastric acid output, resulting in either hypochlorhydria or hyperchlorhydria, which determines the type of peptic ulcer. Hypochlorhydria results from suppressed gastric acid secretion and may lead to pangastritis and the formation of gastric ulcers. These patients have an increased prevalence of corpus atrophy and intestinal metaplasia [28]. On the other hand, approximately 15% of patients infected with *H. pylori* develop hyperchlorhydria with predominant antral gastritis associated with duodenal ulcers. 

### 2.3. Mucosa-Associated Lymphoid Tissue (MALT) Lymphoma

Another gastric condition involving *H. pylori* is extranodal marginal zone gastric mucosa-associated lymphoid tissue (MALT) lymphoma, a non-Hodgkin’s lymphoma, an indolent lymphoproliferative disease involving small heterogeneous B lymphocytes. Gastric MALT lymphoma accounts for 40 to 50% of primary gastric lymphomas and 1 to 6% of all gastric malignancies [29]. Although the stomach is the most common site of MALT lymphomas, it usually lacks organized lymphoid tissue. Still, active chronic inflammation of the *H. pylori*-infected gastric mucosa induces the organization of lymphoid tissue. Studies in vitro showed that gastric MALT lymphoma cells were stimulated by heat-killed *H. pylori* and involved *H. pylori*-specific T cells via CD40 and CD40L interactions [30,31]. Interestingly, in those studies, the investigators noted that the T cell clones from MALT lymphoma had reduced perforin-mediated cytotoxicity and Fas-mediated apoptosis [30]. The gastric mucosa of most cases of gastric MALT lymphoma contains *H. pylori* [32], and eradicating *H. pylori* with the corresponding treatment results in the complete remission of gastric MALT lymphoma in most cases [33,34]. 

### 2.4. Extragastric Manifestations of H. pylori

Although the association of *H. pylori* infection with gastric diseases is well established, the infection is thought to exert systemic pathological effects leading to nongastric clinical outcomes. Those conditions include type 2 diabetes mellitus [35], insulin resistance [36], myocardial infarction [37], iron deficiency anemia [38], primary immune thrombocytopenia [39], and Parkinson disease [40], among others. However, it is unclear how infection with *H. pylori* is positively associated with these non-GI disorders.

As research into *H. pylori* disease associations and efforts to eradicate this common human pathogen have expanded, leading to the increased prevalence of some conditions, a question that has emerged is whether *H. pylori* is a true pathogen or a commensal organism. Various studies have credited *H. pylori* with positive effects for the host because *H. pylori* infection was noted to be inversely associated with the development of some disorders. For instance, there seems to be an inverse relationship between *H. pylori* infection and gastroesophageal reflux disease (GERD) [41]. A cross-sectional case-control study of 5616 subjects undergoing both upper endoscopy and *H. pylori* serology reported an inverse relationship between the presence of *H. pylori* and GERD [41]. In that study, *H. pylori* prevalence was lower in cases with reflux esophagitis than in the controls (38.4% vs. 58.2%, *p* < 0.001). A meta-analysis showed that eradicating *H. pylori* could lead to erosive GERD [42].

Interestingly, a meta-analysis also showed an inverse correlation between *H. pylori* colonization and the risk of esophageal cancer [43]. The study suggested that the increase in esophageal cancer incidence may be linked to the decreased prevalence of *H. pylori* in Western countries. A likely mechanism underlying this outcome is *H. pylori* urease activity (described below in detail), which neutralizes gastric acidity and, in turn, decreases the risk of GERD. 

Other studies have also attributed *H. pylori* colonization with protection against childhood asthma, inflammatory bowel disease (IBD), and celiac disease. The relationship between *H. pylori* and asthma has been the subject of active investigation. Evidence suggests that *H. pylori* infection may be associated with a reduced risk of developing asthma [44,45]. As the incidence of *H. pylori* infection decreases in developed countries and various developing areas, asthma in children and other atopic disorders are rising. Chen and colleagues found a compelling inverse relationship between *H. pylori* infection and the early onset of asthma [44]. A hospital-based case-control study of a pediatric population reported that children who were *H. pylori* seropositive had a reduced likelihood of developing asthma compared to seronegative children (adjusted OR, 0.31 [95% CI, 0.10–0.89]) [46]. *H. pylori* infection may protect from asthma and atopy by promoting an immune response that reduces inflammation in the airways, a crucial feature of asthma. The inverse relationship between *H. pylori* infection and asthma may involve the *H. pylori* induction of a polarized Th1 response [14,47]. Several mechanisms contribute to the prevalent *H. pylori*-induced mucosal Th1 response. One virulence factor discussed below is the *H. pylori* neutrophil-activating protein (HP-NAP), which strongly upregulates both IL-12 and IL-23 production, fostering a polarized Th1 response [48,49]. The resulting cytokines from those Th1 cells may inhibit the Th2 responses characteristic of atopy. Another possible mechanism underlying the inverse association between *H. pylori* and asthma is the induction of regulatory T cells (Tregs) by *H. pylori* infection [50,51], which may influence the prevention of allergic disease. In fact, *H. pylori*-positive persons have higher gastric and circulating Treg levels than *H. pylori*-negative individuals [52,53]. The remote regulation of the respiratory mucosa by immune responses in the gastrointestinal mucosa is consistent with the concept of the “common mucosal immune system” [54].

IBD is another disease noted to have an inverse association with *H. pylori* infection. IBD is an umbrella term for chronic relapsing–remitting digestive disorders, including Crohn’s Disease (CD) and ulcerative colitis (UC). *H. pylori* infection was regarded as a possible IBD risk factor due to similarities in the immunobiology of *H. pylori* infection and IBD. However, multiple studies have suggested an inverse association between *H. pylori* infection and the prevalence of IBD. There are essential epidemiological differences between *H. pylori* and IBD. *H. pylori* infection is more prevalent in developing countries than in developed countries. In contrast, the opposite is true for IBD, which is more prevalent in developed than developing countries. The prevalence of IBD is steadily increasing in developed countries, while rates of *H. pylori* infection are decreasing. IBD is less frequent among individuals who are *H. pylori* seropositive when compared to seronegative subjects [55]. A meta-analysis of 80,789 subjects (6130 patients with IBD and 74,659 non-IBD controls) revealed a significant negative correlation between IBD and *H. pylori* infection [56]. In that study, the investigators observed a consistent negative association between IBD and *H. pylori* infection regardless of age, ethnicity, and detection methods [56]. These observations suggest that *H. pylori* might exert a protective effect against IBD.

## 3. *H. pylori* Virulence Factors and Cancer Mechanisms

*H. pylori* possesses a panel of virulence mechanisms that allow it to survive in the harshest environment of the body, which is the stomach (pH between 1.5 and 3.5) [57], colonize the gastric mucosa, influence the integrity of the epithelium, and subvert the host response to establish persistent infection and induce pathology [58,59,60,61,62,63]. Furthermore, the mechanisms that allow *H. pylori* to promote GC include multiple virulence factors, described below. 

### 3.1. Urease

*H. pylori* urease is a critical virulence factor for *H. pylori*. It is the most abundant protein expressed by *H. pylori*, representing 10–15% of total protein by weight. [64]. *H. pylori* urease is a multimeric nickel-containing enzyme consisting of two subunits, α and β subunits, of 29.5 kDa and 66 kDa, respectively, which catalyzes the hydrolysis of urea into carbonic acid and ammonia. Six α and six β subunits aggregate to form a multimeric complex of 550 kDa. [64]. The released ammonia from urea hydrolysis increases the pH and provides a local protective environment for *H. pylori*. 

*H. pylori* urease can play a role in pathogenesis beyond its enzymatic action. Urease influences the host response by activating monocytes and polymorphonuclear leucocytes, leading to inflammation and damage to the epithelium [65]. Approximately 30% of the total urease is located on the bacterial surface [66,67]. The mechanism behind the presence of urease on the bacterial surface involves bacterial autolysis and association with remaining whole cells [68]. This surface localization allows urease to mediate additional effects via interactions with proteins on the surface of epithelial cells. We showed that urease binds to class II major histocompatibility complex (MHC) and triggers apoptosis of gastric epithelial cells [69,70], which express class II MHC molecules [71,72]. This interaction is mediated by the urease A subunit [73]. Interestingly, the urease B subunit also contributes to the adhesion of *H. pylori* to gastric epithelial cells via association with CD74 [74]. Using recombinant urease subunits and urease B knockout bacteria, *H. pylori* urease B was shown to bind CD74 on gastric epithelial cells and induce NF-κB activation and interleukin-8 (IL-8) production [74]. These responses decreased in the presence of blocking CD74 with monoclonal antibodies. 

An interesting recent report showed that urease may contribute to the carcinogenic potential of *H. pylori* via the induction of hypoxia-induced factor-1α (HIF-1α) [75]. HIF-1 is a transcription factor found as a heterodimer formed by the subunits α and β. The cytoplasmic levels of the α subunit increase under hypoxic conditions. During the development and progression of many forms of cancer, including gastric cancer, HIF-1α is often noted [76]. A noncanonical role of HIF-1α is the reduction in Cyclin D1 half-life and perturbation of the cell cycle [77], and boosting angiogenesis [78]. The investigators showed that this effect of *H. pylori* urease on HIF-1α involved TLR2 activation by urease B, recently found to bind TLR2 [79], and was independent of its enzymatic activity. 

### 3.2. Adhesins

Adhesins are bacterial cell-surface proteins contributing to the bacterial attachment to host cells. The adhesion of *H. pylori* to the gastric epithelium is a key step for colonizing the gastric mucosa. A bacterial adhesin that contributes to this process and determines colonization density, encoded by the *babA2* gene, is the blood group antigen binding adhesin (BabA), which bind to Lewis B blood group antigens (Le^b^) on gastric epithelial cells [80,81]. One study classified *H. pylori* strains into BabA high producers (BabA-H) with Le^b^ binding activity, BabA low producers (BabA-L) without Le^b^ binding activity, and BabA-negative strains (lacking the *babA* gene) [77]. Studies by Sheu et al. noted that strains positive for the *babA2* gene with lower levels of BabA expression appeared to be associated with the highest GC risk [82]. 

The *H. pylori* outer inflammatory protein A (OipA), or HopH, is an outer membrane protein that behaves as an adhesin and stimulates IL-8 secretion. Yamaoka’s group showed that the functional status (on or off) of OipA is controlled by a slipped-strand mispairing mechanism that depends on the number of CT dinucleotide repeats in the 5′ region of the *oipA* gene [83]. *H. pylori* strains with the *oipA* gene on “on” status are associated with greater colonization density, higher IL-8 production, and, consequently, higher neutrophil infiltration. Further, strains with the *oipA* gene on “on” status significantly associate with GC and PUD. HopQ is another outer membrane protein that behaves as an adhesin. HopQ binds to the human carcinoembryonic antigen-related cell adhesion molecules (CEACAMs) on host cells. HopQ alleles are usually found in *H. pylori* strains that contain the *cag* pathogenicity island (PAI, discussed below) [84]. The binding of HopQ to CEACAM enables the translocation of *H. pylori* CagA oncoprotein into epithelial cells [85], and is also important in the regulation of the NF-κB pathway [86,87]. 

The adherence-associated lipoproteins A and B (AlpA/AlpB) are outer membrane proteins participating in *H. pylori* binding to gastric epithelial cells and enhancing colonization. They also induce the production of the inflammatory cytokines IL-6 and IL-8 [88]. Proof of their role in adhesion came from studies in which antiserum against the AlpA fusion protein or knockout of *alpAB* or *alpA* genes blocked or decreased the binding of *H. pylori* to epithelial cells [89,90]. Moreover, we reported that AlpAB mediated binding to live gastric epithelial cells and triggered cellular signaling pathways within the cells [88]. Despite the multiple adhesins employed by *H. pylori* to attach to gastric epithelial cells, the deletion of AlpAB alone reduced bacterial binding by 60–70%, and reduced *H. pylori* colonization of mice. 

Sialic acid-binding adhesin A (SabA) is another crucial virulence factor that facilitates adherence to the gastric epithelium and colonization of the gastric mucosa [91]. SabA is a sizeable outer membrane protein encoded by the *sabA* gene, which is present in the *H. pylori* cag pathogenicity island (described below) [92]. SabA is a pleiotropic protein with sialic acid-binding activity. It links the sialyl-Lewis A and X glycan antigens explicitly. SabA specifically binds to α2-3-linked sialic acids abundantly expressed on the gastric mucosa, especially in the gastric antrum, where *H. pylori* primarily colonize [93,94]. The SabA binding to sialic acids on surface gastric mucins aids *H. pylori* adherence to the epithelium, a crucial step in establishing infection. In addition to sialic acid binding, SabA has been shown to interact with other host molecules, such as laminin [93], fibronectin [93], and blood group antigens. These interactions enhance the ability of *H. pylori* to adhere to diverse types of gastric epithelial cells, and promote colonization in different stomach regions. 

Various allelic forms of these adhesins and OMPs, such as BabA, SabA, OipA, and HopQ, have been linked to GC. For instance, OipA activates epidermal growth factor receptor (EGFR), which sets off AKT and β-catenin cascades, and stimulates cellular proliferation and oncogenic transformation via LEF/TCF transcription factors. The translocation of β-catenin into the nucleus, and subsequent association with LEF/TCF transcription factors, are critical in Wnt signaling, which is triggered abnormally in several cancers [95]. The binding of BabA to Lewis^b^ enhances the pathogenicity of *H. pylori* T4SS (discussed below) and triggers cellular proliferation and expression of proinflammatory cytokines, and oncogenic-related factors [96]. 

### 3.3. CagA and the Pathogenicity Island

The *H. pylori cag* pathogenicity island (*cag* PAI) is a region of about 40 kb DNA that encodes the *cytotoxin-associated gene A* (*cagA*), and approximately 27–30 additional genes that encode proteins that form a type IV secretion system (T4SS) [97]. The T4SS is a syringe-like pilus feature that facilitates the translocation of the effector protein CagA and other bacterial products into gastric epithelial cells, influencing the pathogenesis of *H. pylori*. CagA is a 120–140 kDa protein whose expression by *H. pylori* represents the most potent risk factor for GC [98]. The CagA N-terminal domain contains a binding site for α5β1 integrin, which is a crucial interaction in transferring CagA into the gastric epithelial cells [99,100]. Once inside gastric epithelial cells, CagA binds to phosphatidyl serine (PS) in the inner surface of the cell membrane and is tyrosine-phosphorylated by Src/Abl tyrosine kinases at glutamate-proline-isoleucine-tyrosine-alanine (EPIYA) motifs located in the C-terminus of CagA. Phosphorylated CagA forms complexes with Src-homology 2 (SH2) domains in SHP2, Grb2, and CSK, and modifies several signaling pathways in gastric epithelial cells, resulting in anomalous cytoskeletal changes, cellular proliferation, and differentiation as well as induction of inflammatory cytokines [101]. The magnitude of activation of downstream pathways depends on the type of CagA EPIYA motif and the number of copies. EPIYA motifs vary widely among *H. pylori* strains and are classified into four types based on the variation of flanking regions and orders of spacers. The four EPIYA motif types are EPIYA-A, -B, -C, and -D. CagA in Western strains of *H. pylori* contains EPIYA-A, EPIYA-B, and EPIYA-C motifs. In contrast, CagA from East Asian *H. pylori* strains have EPIYA-A, EPIYA-B, and EPIYA-D, but not the EPIYA-C motif [102]. Higashi and colleagues reported that the EPIYA-C segment comprises 34 amino acid residues that variably repeat [102]. These motifs in the C-terminus of CagA enhance its polymorphism, and are present as tandem repeats varying from one to seven [18]. The levels of phosphorylation and the effects seen in gastric epithelial cells are proportional to the number of EPIYA motifs [103]. Because CagA rarely contains EPIYA-C and EPIYA-D motifs simultaneously, EPIYA-C is considered characteristic of Western CagA, while EPIYA-D is a feature of East Asian CagA. The East Asian CagA has a stronger SHP-2 binding, which confers a stronger ability to perturb cellular functions [102], which may explain differences in GC incidence in East Asia versus Western countries. 

CagA has also been reported to impact epithelial cells in a tyrosine phosphorylation-independent manner. CagA has a conserved repeat motif, responsible for phosphorylation-independent activity (CRPIA: FPLKRHDKVDDLSKVG), in its C-terminal region, different from the EPIYA motifs. The CRPIA motif in nonphosphorylated CagA interacts with c-Met, the hepatocyte growth factor scatter factor receptor, which plays a role in the invasive growth of neoplastic cells. CagA binds c-Met, which then binds phospholipase Cγ (PLCγ) and turns on phosphatidylinositol 3-kinase/Akt signaling. This initiates β-catenin and NF-κB signaling, leading to proliferation and inflammation [104,105]. Growth factor receptor-bound protein 2 (Grb2) also interacts with CagA independent of tyrosine phosphorylation, and this interaction stimulates the Ras/MEK/ERK pathway to cause cell scattering and proliferation [106]. 

Support for the role of CagA as an oncoprotein was obtained from multiple observations. A study by Peek’s group that included Mongolian gerbils infected with the carcinogenic strain 7.13 resulted in gastric dysplasia and cancer in >50% of gerbils infected with the wild-type strain. In that study, none of the gerbils infected with the *cagA***^−^** mutant strain developed these preneoplastic lesions [107]. Various studies in which the *H. pylori cagA* gene alone was transfected into cells have shown that its expression significantly affects the transfected cells. In one report, CagA transfected into human gastric epithelial cells revealed that CagA targets partitioning-defective 1 (PAR1). The CagA–PAR1 interaction causes junctional and polarity defects that release cells from growth-inhibitory signals and promote neoplasia [108]. Using the transfection approach, another group demonstrated that CagA interacted with E-cadherin and perturbed the formation of E-cadherin/β-catenin complexes, leading to the accumulation of β-catenin in the cytoplasm and nucleus, and activating β-catenin signaling [109]. In that study, CagA-transfected cells expressed intestinal-specific molecules as an indication of intestinal transdifferentiation of gastric epithelial cells. Another process whereby CagA promotes gastric cancer was recently reported, in which CagA phosphorylation affects the ubiquitin-proteasome system by binding the E3 ubiquitin ligases SIVA1 and ULF [110]. That interaction caused the activation of ULF, and the degradation of SIVA1 and the tumor suppressor p14ARF. The suppression of ARF results in the inhibition of apoptosis and the oncogenic stress response, advancing cancer. Perhaps the most convincing study was reported by Ohnishi et al., who engineered transgenic mice expressing wild-type or phosphorylation-resistant CagA [111]. The mice expressing wild-type CagA developed gastric epithelial hyperplasia, and some developed gastric polyps and adenocarcinomas. Remarkably, some wild-type CagA transgenic mice developed hematological malignancies such as myeloid leukemias and B-cell lymphomas. In contrast, mice expressing phosphorylation-resistant CagA did not have evidence of these pathological abnormalities [111]. These observations led to CagA being regarded as a bacterial oncoprotein of importance in human neoplasia.

Another bacterial product translocated by the T4SS into gastric epithelial cells is peptidoglycan, which is recognized by NOD1 (Nucleotide Binding Oligomerization Domain Containing 1), a cytoplasmic pattern recognition receptor [112,113,114], abundantly expressed in gastric epithelial cells [114]. NOD1 activation results in NF-κB activation and inflammatory cytokine production [115]. 

### 3.4. Vacuolating Toxin (VacA)

The *H. pylori* vacuolating toxin (VacA) is a major secreted virulence factor without known homologs in other bacterial species except for *H. cetorum*, a Helicobacter species found in the stomachs of marine mammals [116]. The VacA toxin is a pore-forming toxin secreted by *H. pylori,* and has pleiotropic effects on host cells. The pores induced by VacA in the membranes of the gastric epithelial lining lead to the leakage of small molecules and ions from the cells. This damage to the cells disrupts the epithelium’s barrier function, facilitating invasion and colonization of the stomach. VacA can also induce apoptosis in immune cells and gastric epithelial cells. This contributes to gastric inflammation and the progression of *H. pylori*-associated diseases. VacA also modulates the host immune response to *H. pylori* by suppressing the activation of specific immune cells and inhibiting the production of cytokines. This permits *H. pylori* to evade immune-mediated clearance and establish a persistent infection in the stomach.

VacA is first synthesized as a 140 kDa protoxin with an N-terminal signal peptide, a central region representing the toxin, and a C-terminus that mediates transport function. The central region (about 88 kDa), the mature virulent form of the toxin, is secreted after processing and is further cleaved into an A subunit and B subunit of 33 and 55 kDa, respectively. The p33 form was initially regarded as the pore-forming subunit, and the p55 form was ascribed to the cell binding function [117,118]. However, both subunits bind and form vacuoles [119,120]. *H. pylori* VacA binds to sphingomyelin on lipid rafts [121], and is then endocytosed via a clathrin-independent route [122]. VacA is delivered to early endosomes, which are routed by F-actin comets to become integrated into motile vesicles that fuse with mitochondria or late endosomes, where VacA induces apoptosis or vacuolation, respectively [123,124]. VacA has multiple damaging effects on mitochondria, including activation of the proapoptotic proteins Bax and Bak [125], cytochrome c release [126], and mitochondrial fragmentation [127], ultimately resulting in cell death. 

Even though all strains of *H. pylori* have the *vacA* gene, there is substantial diversity in the gene, which includes five heterogenic regions with considerable sequence differences. The first three regions identified correspond to the signal (s) sequence, middle (m), and intermediate (i) [128]. There are two genotypes for s (s1a-c, s2), four for m (m1a, m1b, m1c, and m2), and three for the i region (subdivided into i1a, i1b, and i2). More recently, the deletion d-region (d1 and d2) and the c-region (c1 and c2), which correspond to a 15 bp deletion at the 3′ end of the p55 domain of the *vacA* gene, were described [129,130]. The signal peptide region of VacA provides vacuolating activity and target specificity. The s2 region contains an additional 12-amino acid N-terminal sequence which abolishes the formation of anion-selective channels and cell vacuolation [131]. While s1/m1 and s1/m2 VacA genotypes cause severe chronic inflammation when compared to the other genotypes, *H. pylori* strains with m1 VacA represent a higher risk factor for gastric ulcers [132], and VacA s1 and m1 genotypes are associated with higher pathology, including substantial infiltrates of neutrophils and lymphocytes, gastric atrophy, and intestinal metaplasia [133,134]. Yamaoka’s group showed that the *vacAs1i1m1* allelic combination is strongly associated with the existence of *cagA* [135], and a recent study by Chang and colleagues demonstrated that strains expressing both *vacAs1m1* and *cagA* concurrently have a 4.8-fold greater risk of inducing preneoplastic lesions than other *H. pylori* strains [136]. The i region is found between the s and m regions, and is associated with polymorphism and GC [137]. The m region has a 148 amino acid segment that defines VacA’s cell binding specificity [138]. The roles of the c and d regions in the biology of VacA are currently unknown. 

VacA is crucial in *H. pylori*’s ability to evade host immunity since it can act on several immune cells affecting innate and adaptive immunity. VacA impairs the functions of macrophages [139,140], eosinophils [141], mast cells [142], dendritic cells [143], and lymphocytes [144,145]. Due to its effects on endosomal trafficking, VacA interferes with antigen processing by B cells. Molinari and colleagues showed that VacA disrupts the proteolytic processing of antigens and inhibits the Ii-dependent antigen presentation pathway, which involves newly synthesized class II MHC molecules in endosomes [146]. Macrophages, another professional antigen-presenting cell type, are also disrupted by VacA in their ability to process antigens as the toxin induces the formation of large vesicles termed megasomes, impairing the maturation and function of endosomal compartments [147]. VacA also impairs T cells as it efficiently prevents the proliferation of T cells by inducing a G1/S cell cycle arrest. VacA was shown to block the T cell receptor/interleukin-2 (IL-2) signaling pathway at the Ca^++^-calmodulin-dependent phosphatase calcineurin [145]. These effects of VacA on immune cells undoubtedly contribute to immune avoidance by *H. pylori* and highlight its important role in the pathogenesis of *H. pylori*-associated diseases.

### 3.5. H. pylori Neutrophil Activating Protein (HP-NAP)

HP-NAP, or NapA, is another critical virulence factor of *H. pylori* [49]. HP-NAP is a highly conserved protein encoded by the *napA* gene expressed by most *H. pylori* strains. It is a 164-amino acid protein secreted by *H. pylori* and can be found in both the bacterial cell wall and the extracellular milieu. NAP’s three-dimensional structure was characterized by Zanotti et al. as having a quaternary structure consisting of a spherical dodecamer containing ∼17 kDa identical subunits with a four-helix bundle structure similar to bacterial ferritins [148]. HP-NAP is a multifunctional protein that has been shown to interact with various cells and modulate host immunity.

Among the critical functions of HP-NAP is its ability to activate neutrophils to generate reactive oxygen species (ROS) [149] and reactive nitrogen species (RNS), and to adhere to endothelial cells [150]. ROS and RNS induced by *H. pylori* can cause DNA damage and mutations that lead to the initiation of cancer [151]. HP-NAP activates neutrophils by binding to specific receptors on their surface, such as the formyl peptide receptor 1 (FPR1) and toll-like receptor 2 (TLR2) [48,152], which stimulate the release of ROS, RNS, and inflammatory cytokines [48,153,154,155]. A recent report showed that HP-NAP could induce the formation of neutrophil extracellular traps (NETs) [155], a novel mechanism of neutrophil-mediated host defense [156]. Cytokine production by HP-NAP-activated neutrophils leads to further neutrophil recruitment to the site of infection. The immune responses are meant to clear *H. pylori*. Still, they can also cause collateral damage to the gastric mucosa, leading to inflammation and tissue injury [157], which are outcomes that implicate HP-NAP in GC. One mechanism whereby HP-NAP promotes tissue damage is through stimulating neutrophils to produce high levels of oxygen radicals, which are involved in the pathophysiology of GC [158]. Overall, the chronic inflammation of the gastric mucosa is a pivotal contributor to the development of *H. pylori*-related diseases, including GC.

In addition to its influence on innate immune cells, HP-NAP affects the adaptive immune response. HP-NAP crosses the epithelial barrier and promotes the skewing of CD4^+^ T helper (Th) cell responses to Th1 responses by inducing the production of IL-12 and IL-23 by neutrophils, monocytes, and dendritic cells [48,159]. Those cytokines promote the differentiation of monocytes toward matured dendritic cells (DCs). DCs respond to HP-NAP with increased expression of class II MHC molecules, and produce IL-12 to further the polarization of Th1 cells [159]. It is important to note that IL-23 produced in response to HP-NAP may promote the development of Th17 and the production of interleukin-17 (IL-17). IL-17 is a proinflammatory cytokine critical in the immune response to bacterial infections. IL-17 may contribute to the recruitment of neutrophils and the clearance of *H. pylori* infection. However, excessive production of IL-17 can also contribute to chronic inflammation and tissue damage. 

### 3.6. H. pylori γ-Glutamyltranspeptidase (GGT)

GGT is an enzyme that plays a crucial role in *H. pylori*’s survival and pathogenesis. *H. pylori* GGT (HpGGT) is a type I membrane protein anchored to the bacterial outer membrane. HpGGT is highly conserved and present in all strains [160]. It involves several critical functions, including glutathione detoxification, nutrient acquisition, and host immune response modulation. HpGGT catalyzes transpeptidation and hydrolysis of the gamma-glutamyl group of glutathione and similar compounds [161]. HpGGT hydrolyzes glutamine into glutamate and ammonia, and glutathione into glutamate and cysteinylglycine [162]. Since *H. pylori* cannot take up extracellular glutamine and glutathione directly, this enzyme allows *H. pylori* to use extracellular glutamine and glutathione as sources of glutamate for subsequent use in the tricarboxylic acid cycle [162]. HpGGT is vital for the survival of *H. pylori* in acidic conditions, as *H. pylori ggt*^-^ isogenic mutant strains cannot colonize the gastric mucosa in animal models of infection, or do so less efficiently [163,164]. A study by Chevalier et al. reported that *H. pylori ggt*^-^ mutants could not be recovered from the stomachs of mice at 3–60 days postinfection [164].

HpGGT has wide-ranging effects in gastric epithelial cells and plays a significant role in the pathogenesis of Hp-induced GC. HpGGT functions through multiple pathways to damage the gastric epithelial barrier. HpGGT induces epithelial cell apoptosis by a mitochondria-dependent pathway and reduces cell viability [165]. HpGGT also promotes cell death by reducing survivin levels [166], prompting cell cycle arrest at the G1-S phase transition [167], and increasing ROS production, resulting in glutathione depletion and DNA damage [168]. HpGGT stimulates the expression of heparin-binding epidermal growth factor-like growth factor (HB-EGF), a ligand of the epidermal growth factor receptor (EGFR) [169]. HB-EGF binding to EGFR initiates the Raf/Ras/MEK/Erk and PI3K/Akt pathways, reducing apoptosis and promoting proliferation [170]. The expression of HB-EGF is increased in various types of cancer, including GC [171]. HB-EGF also contributes to GC progression by potentiating the epithelial–mesenchymal transition [172]. *H. pylori* infection also affects mesenchymal stem cells (MSCs) by inducing their migration to the gastric mucosa, where they may contribute to GC development by differentiating into epithelial cells or assisting in angiogenesis [173]. HpGGT was recently shown to disturb MSCs by obstructing alpha-ketoglutarate to boost trimethylation of histones H3K9 and H3K27, triggering PI3K/AKT signaling, and helping proliferation, migration, self-renewal, and pluripotency in cancerogenesis [174]. Another recent report showed that HpGGT might help GC development by activating the Wnt/β-catenin signaling pathway through upregulation of ten-eleven translocation 1 (TET1) [175], which is a crucial DNA demethylase and is overexpressed in GC. 

HpGGT also affects immune cells directly. One of the earliest studies on the immune response effects of HpGGT showed that the enzyme could dampen T cell proliferation [176,177]. It was reported to induce cell cycle arrest at the G1 phase due to interference with the Ras-dependent signaling pathway [177]. This effect of HpGGT on T cell proliferation is thought to mediate immunosuppression, which assists in the persistence of *H. pylori* infection. Another study showed that HpGGT induced microRNA-155 (miR-155) expression in both CCRF-CEM cells (a human T lymphoblast cell line) and primary human peripheral blood mononuclear cells [178]. This response depends on forkhead box P3 (Foxp3), the master regulator of regulatory T cell (Treg) development [179], and requires activation of the cyclic adenosine monophosphate cascade. Treg cells have an immune suppressive activity, often found in the *H. pylori*-infected gastric mucosa [180], promoting higher *H. pylori* colonization and infiltrating tumors [181]. In support of the role of HpGGT in promoting Treg development, mice infected with *ggt-* isogenic *H. pylori* mutant strains were reported to have lower Treg counts than wild-type-infected mice [143]. These observations suggested that HpGGT is influential in regulating the immune system. 

## 4. Immune Checkpoints in *H. pylori* Infection as Immune Escape Mechanisms

In addition to the various immune evasion mechanisms associated with the virulence factors described above, *H. pylori* stealthily directs the expression of multiple receptors that influence T cell activity. The optimal balance between protective immunity and immune tolerance controls immune responses. T cell activity is determined by the combination of signals initiated by T cell receptor (TCR) recognition of antigen/MHC complexes and costimulatory molecules on APCs, divided into costimulatory and coinhibitory molecules (Figure 1). While the costimulatory receptors are critical in initiating immune responses, coinhibitory molecules are essential for avoiding immune-driven pathology, but can also restrain immune-mediated clearance of pathogens. Many pathogens and cancers promote inhibitory interactions via immune checkpoint proteins to evade immune clearance. *H. pylori* induces the expression of gastric epithelial cells of the checkpoint inhibitor B7-H1 (aka PD-L1), which inhibits effector T cells via PD-1 engagement on their surface, and fosters Treg cell development [182,183,184]. Xie and colleagues reported that a mechanism used by *H. pylori* to promote B7-H1 expression by gastric epithelial cells is through the inhibition of miR-152 and miR-200b expression [185]. They showed that these miRNAs targeted B7-H1 mRNA and inhibited the expression of B7-H1 by gastric cancer cells. An interesting study by Wu and colleagues showed that the urease B subunit of *H. pylori* could induce the expression of B7-H1 by bone marrow-derived macrophages after binding to myosin heavy chain 9 (Myh9) on their surface [186]. The interaction of urease B with Myh9 caused activation of mTORC1, which leads to increased expression of B7-H1 on macrophages. This contributes to the overall T cell suppression elicited by the various virulence factors described above (Figure 2). Further, the *H. pylori*-infected gastric epithelial cells simultaneously downregulate their expression of the costimulator B7-H2, thus, not only reducing effector T cell responses, but also altering T cell subset balances by increasing Tregs and decreasing Th17 cell numbers [183,184,187]. This imbalance favors bacterial persistence. We demonstrated that *H. pylori* uses *cag*PAI to induce these effects on gastric epithelial cells, and activates the mTOR/p70 S6 kinase pathway to downregulate B7-H2 expression [182,187]. B7-H1 expression is also increased in GC [188,189,190]. In a subsequent report, we demonstrated that the *H. pylori* T4SS-mediated transfer of CagA and cell wall peptidoglycan (PG) fragment upregulated B7-H3 expression by gastric epithelial cells via activation of the p38MAPK signaling pathway [191]. Thus, *H. pylori* effectively hijacks the regulated expression of coinhibitory molecules to avoid immunity, and may favor immune escape by developing cancer cells through these mechanisms. 

## 5. Conclusions and Future Directions

Over the past 40 years, significant advances have been made in our comprehension of *H. pylori*’s properties since its initial identification as a human pathogen. While infection with *H. pylori* was initially linked with chronic gastritis and the development of peptic ulcer disease, research by multiple laboratories worldwide has established the bacterium as the most potent known risk factor for GC, one of the world’s deadliest cancers.

While antibiotic therapies are available to treat *H. pylori* infection, the high prevalence of the bacterium and the escalation of antibiotic resistance highlights the need for a protective vaccine. Though GC can be a devastating diagnosis, its bacterial etiology makes it a vaccine-preventable form of cancer. *H. pylori* has an extensive array of virulence factors that mediate pathogenesis and could represent potential vaccine targets to prevent the morbidity and mortality associated with the infection and GC. Nevertheless, vaccine development efforts have encountered various challenges. The variability among *H. pylori* strains, and the bacterium’s immune escape mechanisms, create significant obstacles to developing an effective vaccine. Clinical trials of candidate vaccines have yielded mixed or disappointing results.

Although an *H. pylori* vaccine is not currently available, continued research is promising and may eventually lead to an avenue to help decrease the burden of *H. pylori*-associated diseases. Research efforts into immune escape mechanisms used by *H. pylori* and GC, including the induced expression of novel immune checkpoint inhibitors, could yield candidate biomarkers to detect cancer progression in its early stages. In summary, *H. pylori* is the foremost risk factor for developing GC, and understanding how to override the bacterium’s immune evasion strategies is crucial for developing new approaches to treat and prevent the conditions associated with the infection.

## Figures and Tables

**Figure 1 microorganisms-11-01312-f001:**
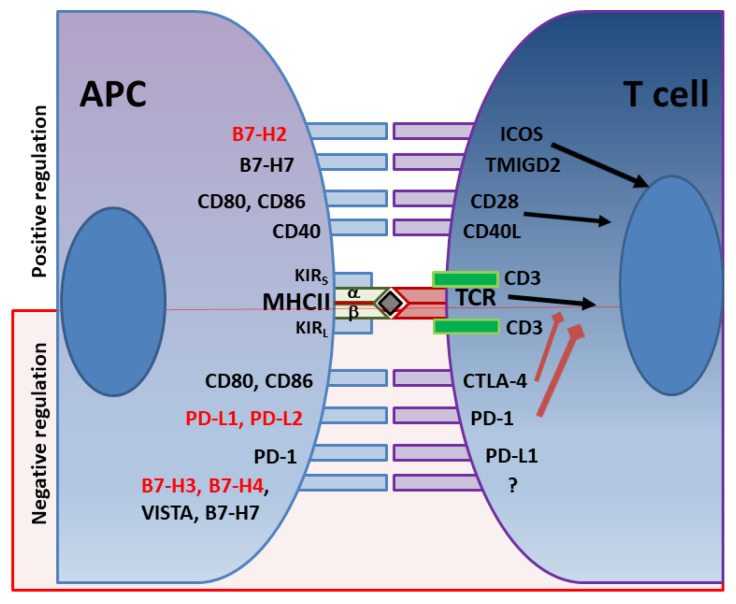
Immune checkpoints regulators of T cell activity. In red are immune checkpoints whose expression is altered by *H. pylori*. Black arrows represent positive signals. Red arrows represent inhibitory signals.

**Figure 2 microorganisms-11-01312-f002:**
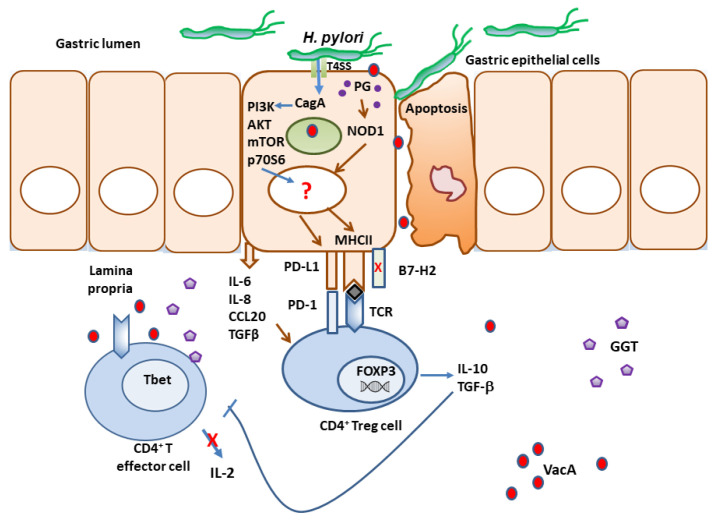
*H. pylori* sets immunosuppressive environment. *H. pylori* interferes with effector T cell functions using an array of virulence factors and manipulating the expression of immune checkpoint regulatory receptors by the epithelium.

## Data Availability

Data sharing is not applicable.

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
