# Peer review of "Helicobacter pylori and Its Role in Gastric Cancer"

_microorganisms, 2023, doi:10.3390/microorganisms11051312_

Round 1

Reviewer 1 Report

The review article “Helicobacter pylori and Its Role in Gastric Cancer” describes the link between Helicobacter pylori infection and the development of gastric cancers. The article discusses the topics of epidemiology, pathophysiology and virulence of this microorganism. The general topic may not be particularly innovative, but the topic is well described and scientifically relevant.

Below, I would like to list some amendments, which if taken into account, will contribute to improving the quality of the manuscript:

- In the title of the manuscript, the name of the bacterial species should be written in lower case, and the entire name of the bacteria should be written in italics.

- Please add a space between the word and the reference number in each sentence.

- Lines 30-34: This sentence is too long. Please make two shorter ones.

- I believe that the paragraph in lines 49-59 should be placed as the second and the paragraph on lines 35-48 as the third. In this way, better connection will be obtained between the introduction and  the next section about stomach diseases.

- Lines 64 and 346: Please remove the extra space.

- Lines 108-109: Please correct the cytokine names to uppercase.

- Please use italics when writing the names of virulence factor genes and bacteria in the lines: 279, 280, 286, 288, 302, 358, 360, 389, 565, 568.

- Please correct the entire list of references in accordance with the requirements of the MDPI.

The article is written correctly in terms of language.

Author Response

I am thankful for the time and thoroughness of the review provided by Reviewer 1. The comments were quite valuable in improving the readability and quality of the manuscript. The comments were all addressed as follows:

  1. “In the title of the manuscript, the name of the bacterial species should be written in lower case, and the entire name of the bacteria should be written in italics.” This correction was made as suggested.
  2. “Please add a space between the word and the reference number in each sentence.” This was corrected in the text.
  3. “Lines 30-34: This sentence is too long. Please make two shorter ones.” This sentence was split into two (Lines 30-34)
  4. “I believe that the paragraph in lines 49-59 should be placed as the second and the paragraph on lines 35-48 as the third. In this way, better connection will be obtained between the introduction and the next section about stomach diseases.” I agree. This is an excellent suggestion and the two paragraphs were rearranged.
  5. “Lines 64 and 346: Please remove the extra space.” This was done. Thanks for pointing it out.
  6. “Lines 108-109: Please correct the cytokine names to uppercase.” This was corrected.
  7. “Please use italics when writing the names of virulence factor genes and bacteria in the lines: 279, 280, 286, 288, 302, 358, 360,389, 565, 568.” You are absolutely correct, and these were all corrected.
  8. “Please correct the entire list of references in accordance with the requirements of the MDPI.” Apparently, the reference software I used failed to list the range of pages and added extra information in other instances. Most of the issues I noticed were electronic publications that do not provide a page range. The entire citation list was reviewed and corrected. A series of additional more recent publications documenting the involvement of virulence factors in cancer was also added.

Once again, thank you for your valuable suggestions.

Reviewer 2 Report

This review covers the important field of Helicobacter pylori infection and human gastric cancer, but the overall goal is not focused enough. A large amount of content (including the discovery history of the bacterium, its relationship with peptic ulcers, immune escape mechanisms, and “Conclusions and Future Directions”) is low in association with gastric cancer. Most of the references are too outdated, and compared to similar published articles, there is not much updated content.

Author Response

First of all, I thank this reviewer for the time and valuable input to enhance the quality of the manuscript. The reviewer indicated in his evaluation that “This review covers the important field of Helicobacter pylori infection and human gastric cancer, but the overall goal is not focused enough. A large amount of content (including the discovery history of the bacterium, its relationship with peptic ulcers, immune escape mechanisms, and “Conclusions and Future Directions”) is low in association with gastric cancer. Most of the references are too outdated, and compared to similar published articles, there is not much-updated content.”

The content regarding history, peptic ulcers, and immune escape is important to mention as it provides some context to the diversity in pathologies associated with the infection. It also brings into focus discussion relevant to the mutually exclusive aspects of duodenal ulcers and gastric cancer. The outdated references are important as they set the stage for the newer aspects of H. pylori’s involvement in gastric cancer. The focus on gastric cancer has been enhanced by including more recent studies listing mechanisms of cancer involvement for each of the virulence factors discussed in the review. New paragraphs with additional citations are included now in lines 273-281, 300-305, 329-337, 497-499, and 578-585. Also, the Conclusions and Future Directions were revised to focus more on gastric cancer. Immune escape mechanisms are an important topic highly relevant to cancer, especially since the incorporation of immune checkpoint inhibitors as part of the cancer immunotherapy armamentarium.
